# Distinguishing the Uterine Artery, the Ureter, and Nerves in Laparoscopic Surgical Images Using Ensembles of Binary Semantic Segmentation Networks

**DOI:** 10.3390/s24092926

**Published:** 2024-05-04

**Authors:** Norbert Serban, David Kupas, Andras Hajdu, Peter Török, Balazs Harangi

**Affiliations:** 1Faculty of Informatics, University of Debrecen, 4028 Debrecen, Hungary; kupas.david@inf.unideb.hu (D.K.); hajdu.andras@inf.unideb.hu (A.H.); harangi.balazs@inf.unideb.hu (B.H.); 2Department of Obstetrics and Gynaecology, Faculty of Medicine, University of Debrecen, 4032 Debrecen, Hungary; petertorokdr@gmail.com

**Keywords:** laparoscopic hysterectomy, semantic segmentation, ensemble model

## Abstract

Performing a minimally invasive surgery comes with a significant advantage regarding rehabilitating the patient after the operation. But it also causes difficulties, mainly for the surgeon or expert who performs the surgical intervention, since only visual information is available and they cannot use their tactile senses during keyhole surgeries. This is the case with laparoscopic hysterectomy since some organs are also difficult to distinguish based on visual information, making laparoscope-based hysterectomy challenging. In this paper, we propose a solution based on semantic segmentation, which can create pixel-accurate predictions of surgical images and differentiate the uterine arteries, ureters, and nerves. We trained three binary semantic segmentation models based on the U-Net architecture with the EfficientNet-b3 encoder; then, we developed two ensemble techniques that enhanced the segmentation performance. Our pixel-wise ensemble examines the segmentation map of the binary networks on the lowest level of pixels. The other algorithm developed is a region-based ensemble technique that takes this examination to a higher level and makes the ensemble based on every connected component detected by the binary segmentation networks. We also introduced and trained a classic multi-class semantic segmentation model as a reference and compared it to the ensemble-based approaches. We used 586 manually annotated images from 38 surgical videos for this research and published this dataset.

## 1. Introduction

The hysterectomy is one of the most frequently performed gynecological surgeries for women [1]. This surgery can be completed in different ways. The traditional techniques comprise vaginal and abdominal hysterectomies [2], but with the improvement of technology over time, the laparoscopic hysterectomy technique was introduced in 1988 [3] to replace the abdominal technique. Although the laparoscopic procedure proved to be superior in terms of postoperative recovery and operative blood loss [4], ureteral injury became a frequent complication of laparoscopic hysterectomies [5,6,7,8].

One of the main reasons for the higher number of ureter injuries is the lack of clear guidelines during a laparoscopic hysterectomy. The risk of injury highly depends on the surgeon’s experience [9]. The technique commonly used to prevent damage to the ureter is etching to the posterior broad ligament and then identifying and moving the ureter laterally [10]. However, this technique can be performed only after the clear identification of the ureter, which is demanding when the surgeon only has visual information available. Apart from this visual information, other regularly used techniques to find the ureter include the placement of a ureteral stent or using x-ray fluoroscopy [11,12]. Nonetheless, these foreign materials lead to additional risks for the patient during the surgery.

In recent years, biomedical image processing has developed rapidly, allowing the application of imaging science to multi-modal biomedical images and providing useful support for examining and diagnosing human patients [13]. Machine learning and artificial intelligence has also boosted the evolution of medical imaging services such as computer-aided diagnosis, image fusion, or image segmentation [14]. Nowadays, these techniques are used in many areas, including in the analysis of CT or MRI images [13], the detection of tumors or cardiac diseases [14], and the registration of multi-modal medical images [15].

### 1.1. Related Works

#### 1.1.1. Ureter and Uterine Artery Detection

As mentioned in the introduction, detecting the ureter and uterine arteries and their differentiation during laparoscopic surgeries is challenging. Still, a few research efforts have been made to address this issue. Accordingly, in our former work [16], we presented a semi-automatic solution to differentiate the ureter from the uterine artery from images captured during laparoscopic surgeries. In this system, a medical expert or assistant manually selected the region of interest (ROI) from each examined image. Then, 224 × 224 sub-images were cut off from the highlighted ROI and classified using GoogLeNet [17]. The final classification of the ROI was decided through majority voting.

The authors of [18] developed and built an image-guided endoscope system using infrared light to detect the ureter during surgery. They proposed a self-adaptive threshold-based segmentation algorithm to enhance the accuracy of ureter detection. Finally, they fused the detection results with the RGB images captured with the endoscopic camera. In [19], a convolutional neural network (CNN) called Kid-Net was proposed to semantically segment arteries, veins, and the ureter. However, this network was trained to segment these structures from volumetric CT images, not the RGB images used in laparoscopic surgeries.

Multi-class semantic segmentation is a reasonable solution for the task of processing images from laparoscopic surgeries. Ref. [20] used the Mask R-CNN [21] to segment multiple instruments from a video of a laparoscopic surgery. Their proposed network uses a region-based CNN to segment the labeled instruments from the images. The work in [22] presented a modified U-Net architecture named U-NetPlus, with the same aim of segmenting the tools used during laparoscopic surgeries. They initiated pre-trained VGG-11 and VGG-16 architectures with batch normalization for the encoder part. In [23], a deep CNN and a dataset named m2caiSeg were provided. This annotated dataset contains labeled organs, instruments, fluids, and arteries. The presented network segments images from laparoscopic surgeries into 19 categories, pixel-wise.

#### 1.1.2. Ensemble Models

Ensemble learning aims to solve a particular machine learning task by combining the knowledge of multiple members. The idea behind this learning method is partly the simulation of human nature, gathering meaningful, diverse knowledge to achieve an enhanced solution to a complex problem [24]. Usually, these members are trained on the same dataset for the same aim but using different approaches. Furthermore, when dealing with a multi-class segmentation problem, binarization may help to solve it. The most common techniques for decomposition are the “one-vs-one” (OVO) [25] and “one-vs-all” (OVA) [26] techniques. The OVO scheme divides the classification problem of m classes into m(m−1)/2 binary classifications. During the training of the binary models, a subset of the original dataset is used, containing only the corresponding classes. Each of these binary models creates a prediction matrix $P_{\left(i,j\right)}\;(x,y)$, where an entry of the matrix indicates the confidence of the classifier for the i-th class over the j-th one for a given image pixel (x,y) with x=1,…,W and y=1,…,H, where the input image has the width W and height H. To aggregate the $P_{\left(i,j\right)}$ matrices for a final prediction, a voting strategy is a simple possibility. Namely, each classifier votes for its predicted class, and the class with the most votes will be suggested finally [27].

On the other hand, the OVA decomposition technique breaks down the m-class classification problem into m binary ones. Usually, the training phase uses the whole dataset, labeling one class as positive and all other classes as negative. Each of these m binary classifiers creates a matrix $P_{i}\;(x,y)$ describing the confidence level of class i for each image pixel (x,y). A standard aggregation scheme for the OVA approach is the Maximum Confidence Strategy, where the class with the highest confidence level will be chosen as the final prediction [27].

The output of an ensemble system can be calculated in many ways. The first and most straightforward way is to aggregate the member outputs by calculating their average, weighted average [28], or arithmetic mean [29]. Another method used to fuse the ensemble members’ outputs is majority voting [30]. This decision model can be boosted by assigning voter confidence levels [31] or weights [32]. Typically, weighted voting ensembles perform better than non-weighted ones since the weights or any other additional parameter let the ensemble be tuned further, which generally leads to a better prediction.

Our work uses the OVA technique to decompose the traditional multi-class U-Net semantic segmentation model into three binary semantic segmentation models. Then, we merge these models to compose both a pixel-wise and a region-based ensemble. We also present a weighted version of these ensembles, where the binary members’ accuracies or confidence levels are considered as weights.

## 2. Materials and Methods

### 2.1. Dataset

The dataset used for this research contains 586 images from 38 surgical videos captured during actual laparoscopic surgeries with a resolution of 1340 × 648 pixels. Each image has a corresponding manually annotated mask, which indicates the pixel-level presence of the ureter, the uterine artery, or nerves on the selected frame. In collaboration with the University of Debrecen, gynecological experts carried out these annotations. For this task, an annotation software was developed, where the gynecologists could draw the boundaries of the occurring ureter, uterine artery, or nerve. Later, these annotated boundaries were filled to create a pixel-level mask of the annotated organs and merged into a single mask to make the final ground-truth mask for the input image. Figure 1 demonstrates the annotation process with its main steps.

From the total of 586 images, we obtained 470 containing uterine arteries, 254 containing the ureter, and 145 containing the nerves. It is also important to note that from these images, 285 frames included both the uterine artery and the ureter. During our research, the dataset was separated into two parts, where 530 images were used to train the semantic segmentation models, and 56 images were separated for testing purposes. During the training of the multi-class and binary networks, 5-fold cross-validation was used, where each split was based on the videos. Therefore, images from the same videos were always kept in the same split. The frames of the test dataset were collected from four surgical videos, which were excluded from the training videos to make our final evaluation reliable. The dataset containing the images from the surgical videos with the corresponding masks (both with separate mask images for every present class and with multi-label mask files well) has been published and is available for further research [33].

### 2.2. U-Net Deep Convolutional Network

The reference model for this task was a multi-class U-Net semantic segmentation model [34]. This model takes the whole dataset with multi-class labeling (0—background; 1—uterine artery; 2—ureter; 3—nerve) and returns a pixel-level prediction map for each image. The U-Net architecture can take a pre-trained encoder and use it during training. U-Net architecture is widely used for semantic segmentation tasks, since the architecture can be modified and improved through different operations quite easily, which can lead to new network architectures based on U-Net such as those seen in [35,36]. For our research, the EfficientNet-b3 [37] encoder was selected and used for the multi-class and binary segmentation networks in this case. Figure 2 illustrates the architecture of the used U-Net semantic segmentation network with the EfficientNet backbone. As this figure demonstrates, the downsampling part of the network represents an EfficientNet-b3 encoder, which uses a 5-staged downsampling method plus an additional downsampling layer containing a Leaky ReLU, a 2 × 2 Max-Pooling layer, and a Dropout layer as well. The upsampling is performed via the default U-Net operations using upsampling layers with transposed 2D convolutional operations. Each final upsampling layer is a 1 × 1 convolution layer attached with a Sigmoid or SoftMax layer, depending on the number of output classes.

As shown in the discussion of the performance of this network in the Results section, this multi-class network could not efficiently find the differences between the ureter, the uterine artery, and nerves. This poor performance was achieved because of the nature of the dataset and the visually similar classes. The distribution of classes in the dataset is highly imbalanced; images with the ureter or uterine arteries are over-represented compared to those with nerves. The problem with this imbalanced distribution is that even with a higher batch size, it is possible that not all the classes were represented in the batch. This resulted in confusion about the loss function and the correct convergence to its minimum.

### 2.3. Binary Semantic Segmentation Models and Their Ensembles

Due to the abovementioned problems, the multi-class segmentation model had to be decomposed into three separate binary semantic segmentation models. So, instead of having one model which has the goal of learning to segment the ureter, uterine artery, and nerves at the same time, the idea was to create separate models and train each of them to only have the capability to segment one of the three organs. Afterward, these binary segmentation models were merged to predict the existence of the classes on images that could contain any of these classes. All of the binary models, like the original multi-class model, used the U-Net architecture with the EfficientNet encoder. This ensemble of binary models was required to build the ability to predict multi-class images accurately. This section will discuss four possible techniques to build ensemble models that can be applied to our task. 

#### 2.3.1. Pixel-Wise Ensemble

The first approach uses a pixel-wise ensemble of the models. In this ensemble method, each binary model creates a prediction for the input image of size W × H, resulting in three distinct prediction maps $P_{i}\left(x,y\right)$: x=1,…, W; y=1,…, H; and i=1,…, 3, representing the artery, ureter, and nerve segmentations, respectively. Then, the merging algorithm checks every pixel separately in each of these maps. Here, the algorithm has three choices. Firstly, when each binary model predicts the pixel as the background, the final prediction will also be the background class. Secondly, when one of the models predicts only the pixel as positive, it is classified into the corresponding organ’s class. The third scenario is when at least two models predict the same pixel as positive. In this case, the class prediction with the highest confidence level is accepted as the final class for that pixel. That is, this ensemble algorithm provides a final labeled output image $P\left(x,y\right)\in \left\{0,1,2,3\right\}$, x=1,…, W, y=1,…, H, from the binary segmentation members’ prediction matrices $P_{i}\left(x,y\right),\;i=1,\ldots ,3$, as follows:(1)$$P\left(x,y\right)=\left\{\begin{array}{c} 0,\;if\;{\forall iP}_{i}\left(x,y\right)<0.5,\\ \;\underset{i\in \{1,2,3\}}{\mathrm{argmax}}\left(P_{i}\left(x,y\right)\right),\;otherwise. \end{array}\right.$$

Notice that the label 0 in $P\left(x,y\right)$ corresponds to the background class, and the further limitation in (1) that only pixels above the confidence level 0.5 are considered as positive, that is, accepted to correspond to one of the investigated organs. Figure 3 illustrates the results of the pixel-wise ensemble technique. As it can be seen in this figure, the input image flows through the binary segmentation networks that were trained for separate organs, providing three segmentation maps. Afterwards, these maps are fed into the above-mentioned algorithm and equation, which results in the final segmentation map as an output image.

#### 2.3.2. Weighted Pixel-Wise Ensemble

The ensemble technique mentioned above was improved by weighing the member binary models with their respective accuracies $w_{1}$, $w_{2}$, and $w_{3}$. These accuracy values indicated how well each binary member of the ensemble could predict its appropriate class. Using the former notations, the decision model given in (1) can be simply updated to incorporate this weighting approach as
(2)$$wP\left(x,y\right)=\left\{\begin{array}{c} 0,\;if\;{\forall iw_{i}P}_{i}\left(x,y\right)<0.5,\\ \;\underset{i\in \{1,2,3\}}{\mathrm{argmax}}\left({w_{i}P}_{i}\left(x,y\right)\right),\;otherwise. \end{array}\right.$$

#### 2.3.3. Region-Based Ensemble

The region-based ensemble method was added as an extra step to the previously discussed pixel-wise ensemble. The first steps of the algorithm remained the same by selecting the final prediction for each pixel. To ensure that the final prediction map does not contain inconsistent regions within the classified organs, this ensemble algorithm examines every connected component in it.

Namely, we took all of the connected components separately and determined their final labels. We let $R=[r_{k}]$ denote the list of all $r_{k}$ connected components of the prediction matrix $P\left(x,y\right)$ (or $WP\left(x,y\right)$), noting that the cardinality of R (denoted by |R|) may vary image by image since the investigated organs can appear as multiple connected components. Then, the algorithm iterated through all of these $r_{k}$, k=1,…,|R| regions to re-label all of their pixels ${(x^{\prime },y^{\prime })\in r}_{k}$ with the most dominant labels they enclosed, as follows:(3)$$RP(x^{\prime },y^{\prime })=\underset{i\in \{1,2,3\}}{\mathrm{argmax}}\left({|\{(x^{\prime },y^{\prime })\in r}_{k}:P\left(x^{\prime },y^{\prime }\right)=i\}|\right).$$

That is, each $r_{k}$ was re-labeled to the most frequent organ label it contained. Figure 4 depicts a flowchart of this region-based ensemble technique for an example input image. From the results of the pixel-wise ensemble process, it created a binary map of connected components; then, the equation above was used to examine the majority class of each region, which provided the final classes for all separable regions.

#### 2.3.4. Weighted Region-Based Ensemble

Furthermore, similarly to the pixel-wise ensemble method, a weighted version can also be introduced for the region-wise one. For this purpose, using the former notations, the prediction matrix of the weighted pixel-base ensemble $wP\left(x,y\right)$ can be re-labeled to that of the weighted region-based ensemble, as follows:(4)$$wRP(x^{\prime },y^{\prime })=\underset{i\in \{1,2,3\}}{\mathrm{argmax}}\left({|\{(x^{\prime },y^{\prime })\in r}_{k}:wP\left(x^{\prime },y^{\prime }\right)=i\}|\right).$$

### 2.4. Model Evaluation

We used two metrics to measure each model’s segmentation accuracy in our application, both globally at the image level and at the class level. This overall evaluation included the multi-class segmentation model (see Section 2.2), the binary semantic segmentation ones, and the pixel-wise/region-wise, non-weighted/weighted ensemble variants. The first applied metric was the Jaccard score [38], also known as the Intersection over Union (IoU), and this was calculated for each segmentation technique at the image level as
(5)$$Jacard\left(gt,P\right)=\frac{\left|gt\cap P\right|}{\left|gt\cup P\right|},$$
where gt stands for the manually drawn ground-truth mask of the input image and P denotes the prediction mask of the given segmentation model. Since the image backgrounds were quite large, they could have distorted the segmentation score. Thus, we also calculated the score at the class level, restricting to the corresponding organ only via
(6)$${Jaccard}_{c}\left({gt}_{c},P_{c}\right)=\frac{\left|{gt}_{c}\cap P_{c}\right|}{\left|{gt}_{c}\cup P_{c}\right|},$$
where ${gt}_{c}$ and $P_{c}$ are the masks of the ground truth and the segmentation algorithm restricted to the corresponding organ. By following our previous labeling, we also wrote ${Jaccard}_{1}$, ${Jaccard}_{2}$, and ${Jaccard}_{3}$ for the ureter artery, ureter, and nerve classes, respectively. Similar to the Jaccard score, we also calculated the Dice coefficient for all of the segmentation methods for both the image and class levels, as follows: (7)$$Dice\left(gt,P\right)=\frac{\left|gt\cap P\right|}{\left|gt|+|P\right|}$$
(8)$${Dice}_{c}\left({gt}_{c},P_{c}\right)=\frac{\left|{gt}_{c}\cap P_{c}\right|}{\left|{gt}_{c}\right|+|P_{c}|}.$$

Moreover, we used the ${Dice}_{1}$, ${Dice}_{2}$, and ${Dice}_{3}$ notations when we directly wanted to refer to the ureter artery, ureter, and nerves. We also noted that the number of images in the dataset was insufficient to train the network with multi-class segmentation properly. To overcome this problem, the images were augmented by cropping a total of 24 sub-images with a size of 512 × 512 pixels from each. This slicing algorithm randomly selected sub-images from the original one with a special condition. Namely, the selected sub-images needed to contain at least 1/3 of the labeled region. After performing the slicing algorithm on every image, the dataset expanded from its original size of 586 images to 14,100 images. The test dataset was kept in its original form, and all models were evaluated using the original test dataset of 56 images.

## 3. Results

This section summarizes and compares the performance of the proposed segmentation models. For the implementation of this research, we employed Python 3.9 using Tensorflow 2.12.0 and Keras 2.12.0 machine learning frameworks. The training and evaluation of the semantic segmentation models and its ensembles were performed using a NVIDIA Tesla K80 graphics card with 12 GB of memory. During our research, we tested several hyperparameters for training with the learning rate varying from $10^{-3}$ to $10^{-4}$ with five steps. At the end, the best results were achieved using a learning rate of $5\times 10^{-4}$ with a batch size of 16. Due to the lack of computation capacity, this was the maximum batch size we could test for that size of images. For each case, we used the Adam optimizer and trained the network until the validation loss converged to its minimum.

### 3.1. Multi-Class Semantic Segmentation Model

According to the evaluation of the test dataset, the original multi-class model does not show the capability to distinguish the essential regions. The scores corresponding to the class-level performance demonstrate well the inability of the model to differentiate the investigated structures. The only class where this model showed signs of having a predicting capability was the ureter (class 2), where it achieved much higher scores than those it achieved for the other two structures. Still, we can use these multi-class U-Net semantic segmentation results as a benchmark to conduct comparisons with the performances of the ensemble models later.

### 3.2. Decomposed Binary Models

The binary semantic segmentation models were trained five times each by splitting the dataset into five portions. The annotations for the binary segmentation networks were binarized regarding the class that the binary networks were trained for. In every training cycle, different parts of the dataset were used as validation datasets to fine-tune the model. The final evaluation of the binary semantic segmentation models was conducted on the test dataset. At both the image and class levels, we observed certain improvements regarding the multi-class semantic segmentation model.

### 3.3. Ensemble Models

The binary segmentation models were merged into a multi-class semantic segmentation model using the ensemble creation techniques discussed in Section 2.3. If we compare the ensemble techniques with each other, the pixel-wise technique has the lowest accuracy since it creates its predictions with the highest noise over the images. By applying weights, this noise reduces, which leads to a higher segmentation accuracy for the separate classes. By using the region-based ensemble technique, each connected region is assigned with a single label. This removes the inconsistent predictions within the regions, which enhances the performance by a significant margin. The summarized results for all tested models and ensembles are demonstrated in Table 1. If we take a closer look at the performance metrics, the ensemble approach outperforms the classic multi-class segmentation network with a large margin. For the task of uterine artery segmentation, our ensemble techniques scored 0.2260 more for the Jaccard scores and 0.2632 more for Dice on average. The same significant improvement can be seen for the nerve segmentation, with higher average Jaccard and Dice scores by 0.4834 and 0.5, respectively. The multi-class segmentation network achieved the best performance in ureter segmentation; however, the ensemble segmentation networks still made an improvement of 0.0221 for the Jaccard score and performed slightly worse for the Dice score, with a 0.0207 lower average score. These figures indicate that the overall segmentation performance improved a lot, with 0.1614 for the Jaccard score and 0.1051 for the Dice score demonstrating the improvement made by the ensemble segmentation techniques. Figure 5 visualizes the performance improvements achieved for the different organ detection tasks.

As part of our evaluation, we conducted a visual inspection to compare the ensemble models’ segmentation results with the test images’ ground-truth masks. Figure 6 illustrates how well the outputs of the ensembles fit after superimposing them onto the original RGB images. In these examples, we used the following color mapping: uterine artery—red; ureter—green; nerves—blue.

## 4. Discussion

In the previous sections, we presented an evaluation of our proposed ensemble techniques. Our work focused on capturing the differences between three different organs. Using decomposed segmentation networks was an efficient way to carry out this work since we could train each network to learn the features of the separate organs and try to differentiate them from all of the other organs as well. However, this process is harsh on time and computational complexity, which can be considered a weakness of the proposed solution since the evaluation contains four separate steps: the prediction of the three binary segmentation network and the ensemble method. As the ensemble of the multiple segmentation networks is only possible after the prediction of all separate networks, the time consumption is equivalent to that of three basic U-Net segmentation networks, which can be major setback for real-time segmentation. In our future work, we need to optimize this process to make the segmentation more efficient. However, the ensemble algorithms do not contain complex computational operations, so if the predictions of the binary networks are optimized or parallelized, the time consumption of the whole segmentation package can be reduced significantly.

The dataset we used for this work is unique for semantic segmentation tasks based on gynecological surgeries. This is one reason why our proposed solution was evaluated on this dataset alone. This may result in a bias in the trained segmentation models as they were only trained on this single dataset. However, the dataset used contains 38 different examinations, which can help the generalization of the models as they are able to see the organs with enough deviation in terms of shape, texture, or color.

## 5. Conclusions

In this work, we propose a method for semantically segmenting human organs during laparoscopic surgeries. In such datasets as the presented one, the original type of multi-class segmentation model fails to predict each represented structure accurately. To overcome this problem, binary semantic segmentation models were trained to distinguish one dedicated class from all others. Finally, these models were ensembled using two techniques that outperformed the original multi-class network by a large margin. Expanding the dataset could also make the multi-class segmentation model work. Still, since the annotation of surgery images is time-consuming and can only be performed by experts, other methods must be considered, like the presented one. Furthermore, more surgical images would enhance the performance of these binary segmentation models as well. The other main area for improvement in this current work involves lowering the time consumption of the segmentation models. The current research focused on image data; however, optimizing our method to be able to handle image sequences or videos of surgical experiments would make our solution a great benchmark for real-time decision support systems for hysterectomies. However, creating a real-time analysis system takes a lot of effort in terms of optimization logic, and implementation as well. One of the ways to achieve this could be through the fusion of binary models, so that the ensemble is effectively completed within a single segmentation model, which should reduce the computation time by a significant amount.

## Figures and Tables

**Figure 1 sensors-24-02926-f001:**
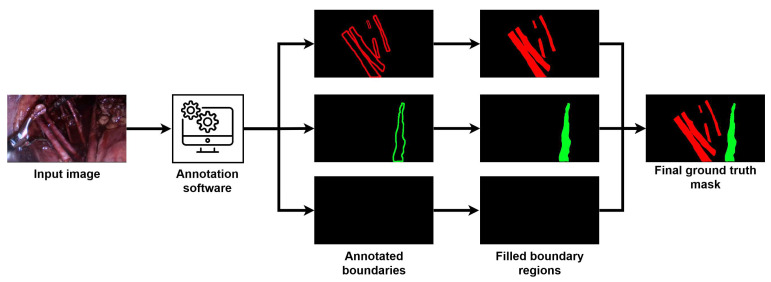
Annotation process for the frames extracted from surgical videos.

**Figure 2 sensors-24-02926-f002:**
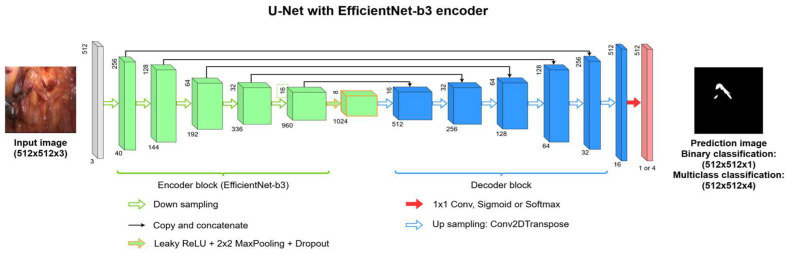
The architecture of the U-Net network with the EfficientNet-b3 encoder.

**Figure 3 sensors-24-02926-f003:**
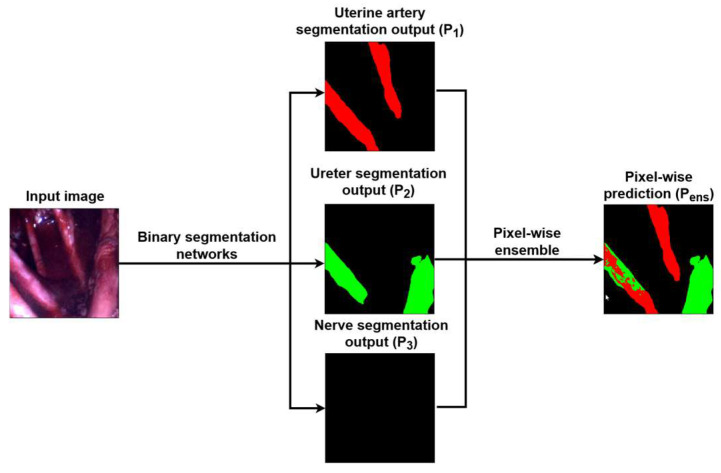
Flowchart of the pixel-wise ensemble technique.

**Figure 4 sensors-24-02926-f004:**
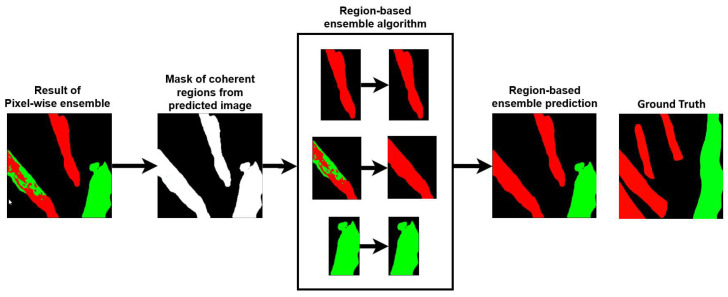
Flowchart of the region-based ensemble technique.

**Figure 5 sensors-24-02926-f005:**
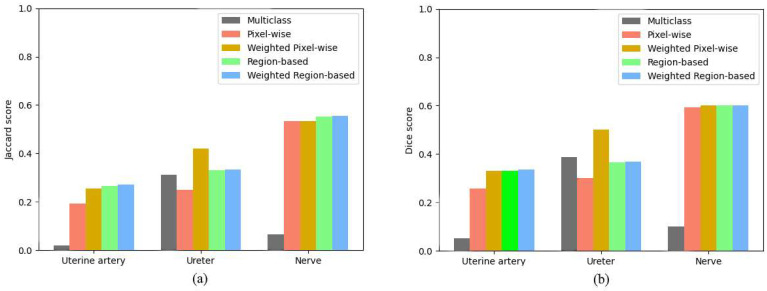
Comparison between the reference multi-class semantic segmentation method and the proposed ensemble methods according to the Jaccard (**a**) and Dice (**b**) scores.

**Figure 6 sensors-24-02926-f006:**
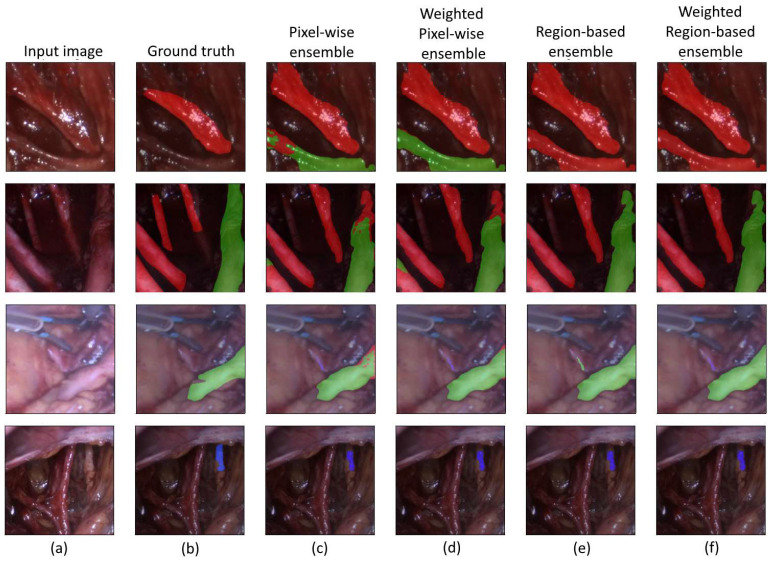
Visual inspection of the segmentation results: input image (**a**); ground truth (**b**); pixel-wise ensemble (**c**); weighted pixel-wise ensemble (**d**); region-based ensemble (**e**); weighted region-based ensemble (**f**); uterine artery—red; ureter—green; nerve—blue.

**Table 1 sensors-24-02926-t001:** Performance of the different models and ensembles on the test dataset.

	Jaccard_1_	Jaccard_2_	Jaccard_3_	Dice_1_	Dice_2_	Dice_3_	Jaccard	Dice
Multi-class U-Net	0.0206	0.3111	0.0653	0.0514	0.3886	0.0997	0.7096	0.8252
Binary U-Net model for the ureter class	0.2272 ± 0.0173	-	-	0.3028 ± 0.0108	-	-	0.8192 ± 0.0071	0.8928 ± 0.0041
Binary U-Net model for the uterine artery class	-	0.3190 ± 0.0849	-	-	0.3880 ± 0.0822	-	0.8565 ± 0.0261	0.9189 ± 0.0158
Binary U-Net model for the nerve class	-	-	0.1262 ± 0.0955	-	-	0.1702 ± 0.1023	0.8173 ± 0.0690	0.8972 ± 0.0438
Pixel-wise ensemble	0.1922	0.2503	0.5332	0.2583	0.3012	0.5936	0.8731	0.9272
Weighted pixel-wise ensemble	0.2562	**0.4201**	0.5533	0.3310	**0.4999**	0.6010	**0.8883**	**0.9358**
Region-based ensemble	0.2663	0.3300	0.5536	0.3316	0.3645	0.6019	0.8461	0.9290
Weighted region-based ensemble	**0.2718**	0.3325	**0.5548**	**0.3372**	0.3681	**0.6025**	0.8764	0.9292

## Data Availability

Data are contained within the article.

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
