# Peer review of "Distinguishing the Uterine Artery, the Ureter, and Nerves in Laparoscopic Surgical Images Using Ensembles of Binary Semantic Segmentation Networks"

_sensors, 2024, doi:10.3390/s24092926_

Round 1

Reviewer 1 Report

Comments and Suggestions for Authors

The manuscript proposes a solution for distinguishing the uterine artery, ureter, and nerve from laparoscopic surgical images using ensembles of binary semantic segmentation networks. The motivation stems from the challenges faced during laparoscopic hysterectomy, where organs are difficult to distinguish visually, impacting surgical precision. The authors utilize semantic segmentation to generate pixel-accurate predictions and compare the performance of ensemble-based approaches with a classic multi-class semantic segmentation model.

Leveraging ensemble-based binary semantic segmentation networks represents an innovative approach to address the problem at hand. By combining multiple models, the proposed method may offer improved accuracy and robustness in distinguishing target structures. The authors also have contributed to the research community by publishing a dataset consisting of manually annotated surgical images. This dataset not only facilitates reproducibility of the study but also serves as a valuable resource for future research in laparoscopic surgery and medical image analysis.

Please some suggestions for minor revision:

The manuscript would benefit from providing more detailed descriptions of the methodology, including the specific architectures of the U-Net models used, training procedures, and ensemble techniques employed. Additionally, it is suggested to introduce more relevant work on the application of U-Net and its variants for semantic segmentation, such as: DOI: 10.1080/17452759.2024.2325572

While the manuscript mentions comparing ensemble-based approaches with a classic multi-class semantic segmentation model, further details on the evaluation metrics used and the comparative analysis results would strengthen the validity of the proposed solution. Providing quantitative performance metrics and discussing the advantages of ensemble methods over the baseline model would enhance the manuscript's impact.

The manuscript presents an innovative approach to address a clinically relevant challenge in laparoscopic surgery using ensemble-based binary semantic segmentation networks. With revisions addressing the suggestions provided, the manuscript has the potential to make a valuable contribution to the field of medical image analysis and laparoscopic surgery. Therefore, I recommend the manuscript for acceptance pending minor revisions.

Comments on the Quality of English Language

English language used throughout the manuscript is very good.

Reviewer 2 Report

Comments and Suggestions for Authors

The paper examines the quality of semantic segmentation in a certain task. It turns out that a single multi-class classifier (commonly adopted approach) is inferior to somewhat more detailed treatment using several one-versus-all classifiers. The payoff is the computational complexity that is higher for proposed approach. All this is not new, however, the paper presents a clear illustration for given case and deserves publishing. The following issues should be addressed.

1. The introductory part of the abstract currently occupies almost half of it, that is too long. The abstract should be more focused on the authors' work. Maybe some more details of the study can be added for balance.

2. In conclusion the authors honestly declare their method is slower than the rivals. However I think more elaboration is needed here, including theoretical and numerical evaluation of the methods' complexities.
